# Synthesis of New Cobalt(III) *Meso*-Porphyrin Complex, Photochemical, X-ray Diffraction, and Electrical Properties for Photovoltaic Cells

**DOI:** 10.3390/molecules27248866

**Published:** 2022-12-13

**Authors:** Soumaya Nasri, Mouhieddinne Guergueb, Jihed Brahmi, Youssef O. Al-Ghamdi, Florian Molton, Frédérique Loiseau, Ilona Turowska-Tyrk, Habib Nasri

**Affiliations:** 1Department of Chemistry, College of Science Al-Zulfi, Majmaah University, Majmaah 11952, Saudi Arabia; 2Laboratory of Physical Chemistry of Materials, Faculty of Sciences of Monastir, University of Monastir, Avenue de L’environnement, Monastir 5019, Tunisia; 3Département de Chimie Moléculaire, 301 rue de la Chimie, Université Grenoble Alpes, CS 40700, CEDEX 9, 38058 Grenoble, France; 4Faculty of Chemistry, Wrocław University of Science and Technology, 27 Wybrzeże Wyspiańskiego, 50-370 Wrocław, Poland

**Keywords:** cobalt porphyrin, synthesis, photophysical properties, X-ray molecular structure, dye degradation, electrical parameters

## Abstract

The present work describes the preparation and characterization of a new cobalt(III) porphyrin coordination compound named (chlorido)(nicotinoylchloride)[*meso*-tetra(*para*-chlorophenyl)porphyrinato]cobalt(III) dichloromethane monosolvate with the formula [Co^III^(TClPP)Cl(NTC)]·CH_2_Cl_2_ (**4**). The single-crystal X-ray molecular structure of **4** shows very important ruffling and waving distortions of the porphyrin macrocycle. The Soret and Q absorption bands of **4** are very red-shifted as a consequence of the very distorted porphyrin core. This coordination compound was also studied by fluorescence and cyclic voltammetry. The efficiency of our four porphyrinic compounds—the H_2_TClPP (**1**) free-base porphyrin, the [Co^II^(TClPP)] (**2**) and [Co^III^(TClPP)Cl] (**3**) starting materials, and the new Co(III) metalloporphyrin [Co^III^(TClPP)Cl(NTC)]·CH_2_Cl_2_ (**4**)—as catalysts in the photochemical degradation was tested on malachite green (MG) dye. The current voltage of complexes **3** and **4** was also studied. Electrical parameters, including the saturation current density (J_s_) and barrier height (ϕ_b_), were measured.

## 1. Introduction

Cobalt metalloporphyrin complexes have been extensively investigated since the early sixties, firstly as biomimetic models of hemoproteins, especially of vitamin B_12_ [1,2,3]. These coordination compounds are indeed explored in many domains, such as catalysts [4], gas detection [5], biological activities, antifungal and antibacterial [6,7], building blocks [8,9], photovoltaic cells [10,11,12], and sensors [13,14]. Organic dyes are used in many industrial applications: paper, cosmetics, but especially in the food, textile, pharmaceutical, and medical diagnostic industries [15,16]. Their disposal is one of the main problems in liquid waste treatment.

The majority of organic dyes are toxic causing serious environmental and human health problems because of their mutagenic [17], teratogenic [18] and carcinogenic properties [19]. Among methods used for the decolorization of dyes, heterogeneous photocatalysis is one of the most used [20,21,22,23,24].

This method is based on the irradiation of a catalyst, usually a semiconductor, which can be photoexcited to create electron-donating or -accepting sites and thus cause redox reactions. If the absorbed photons have an energy greater than the energy difference between the valence and the conduction bands, electron–hole pairs are formed in the semiconductor (holes in the BV band and electrons in the BC band). Notably, porphyrin complexes have been successfully tested as catalysts in the chemical and heterogeneous photocatalysis degradation of organic dyes [25,26,27].

In continuation of our general investigation concerning synthesis, the spectroscopic, electrochemical, and structural characterization of new porphyrin coordination compounds, we recently published several papers concerning zinc(II), magnesium(II) and cobalt(II) metalloporphyrins, which were tested as catalysts in the decomposition of several organic dyes [28,29,30].

To get new insights into the *para*-chloro-substituted *meso*-tetraphenylporphyrin (H_2_TClPP) and the sterically hindered nicotinoyl chloride ligand on the electronic and structural properties of a cobalt coordination compound, the reaction of an excess of nicotinoyl chloride with the (chloride)[*meso*-tetra(*para*-chlorophenyl)porphyrinato]cobalt(III) complex ([Co^III^(TClPP)Cl]) was carried out in dichloromethane under air. The resulting (chlorido)(nicotinoyl chloride)[*meso*-tetra(*para*-chlorophenyl)porphyrinato]cobalt(III) dichloromethane monosolvate with the formula [Co^III^(TClPP)Cl(NTC)]·CH_2_Cl_2_ (**4**) was characterized by UV-vis, IR, ^1^H NMR and fluorescence spectroscopy. Cyclic voltammetry data as well as the X-ray molecular structure and the Hirshfeld surfaces analysis of **4** were investigated. A photodegradation investigation on malachite green (MG) dye using the free-base H_2_TClPP (**1**), the [Co^II^(TClPP)] (**2**) and [Co^III^(TPP)Cl] (**3**) starting materials and complex **4** is reported.

## 2. Experimental

### 2.1. Cyclic Voltammetry Experiments

These experiments were measured using a CH-660B potentiostat. All analyses were done under an argon atmosphere at room temperature in a standard three-electrode electrochemical cell. Tetrabutyl ammonium hexafluorophosphate (TBAPF_6_) was used as the supporting electrolyte (0.1 M) in dichloromethane. The Ag/AgNO_3_ (TBAPF_6_ 0.2 m in CH_2_Cl_2_) redox couple was used as the reference electrode. All potentials quoted in the text were transformed into values for the saturated calomel electrode (SCE) by applying the following equation: E(SCE) = E(Ag/AgNO_3_) + 298 mV.

### 2.2. The Catalytic Degradation

The photocatalytic degradation of the malachite green (MG) dye in the presence of compounds **1**–**4** was realized in air. A 5 mg quantity of H_2_TClPP (**1**) or [Co^II^(TClPP)] (**2**) or [Co^III^(TClPP)Cl] (**3**) or [Co^III^(TClPP)Cl(NTC)]^∙^CH_2_Cl_2_ (**4**) (0.0066, 0.0061, 0.0046 and 0.0046 mmol, respectively) was added to 50 mL of an aqueous solution of MG (20 mg·L^−1^, 0.055 mmol). The pH of the solution was 7. The mixture was first stirred in the darkness for 30 min of illumination to assure that the adsorption–desorption equilibrium was formed. A 300 W Xe lamp was used as the light source with an optical filter (λ > 400 nm) held at about 5 cm from the liquid surface for 30 min. At selected time intervals, an adequate quantity of suspension was centrifuged and filtered using a membrane filter to eliminate solid particles and recover the filtrate for further processing. The degradation efficiency (R%) was measured according to the relation $R\left(\%\right)=\left(\frac{C_{o}-C_{t}}{C_{o}}\right)\times 100$, where *C_t_* and *C*_o_ are the concentrations at time t and starting concentration, respectively.

### 2.3. Synthetic Procedures

All reagents and solvents were obtained commercially and used without further purifications.

In Figure 1 are illustrated the structures of our four porphyrinic compounds **1**–**4**:

#### 2.3.1. Synthesis of H_2_TClPP (**1**)

One gram of 4-chlorobenzaldheyde (0.007 mol) was dissolved using a magnetic stirrer in 100 mL of propionic acid and heated to 120 °C while maintaining stirring. Pyrrole (0.5 mL, 0.007 mol) was then added dropwise to the yellowish solution and the mixture was kept at 120 °C for an additional hour. The resulting solution was cooled, and the tar mixture was filtered to obtain a blue-black precipitate that was decanted with water (5 × 50 mL) and *n*-hexane (5 × 50 mL) and finally dried under vacuum. The crude solid was dissolved in 50 mL of chloroform and purified by silica gel column chromatography using the chloroform as eluent to afford the *meso*-tetra(*para*-chlorophenyl)porphyrin (H_2_TClPP) (**1**) with a yield of 21%.

Anal (%) calcd for C_44_H_26_N_4_Cl_4_ (750.09): C, 70.46; H, 3.49; N, 7.47; found: C, 70.23; H, 3.65; N, 7.62; UV-vis [CH_2_Cl_2_]: λ_max_ (ε.10^−3^ M^−1^.cm^−1^): 421(335), 522(85), 557(56), 599(29), 651(36); ^1^H NMR (400 MHz, CDCl_3_) δ(ppm), 8.89 (s, 8 H β-pyrrole), 8.08 (s, 8H ArH) 7.74(s,8H, ArH), −2.75 (NH); FTIR (solid, $\bar{\nu }$, cm^−1^); 3314 ν[(NH) porphyrin], 2928 ν[(CH) porphyrin], 1470 ν[(C=N),(C=C) porphyrin], 963 [δ(CCH) porphyrin].

#### 2.3.2. Synthesis of [Co^II^(TClPP)] (**2**)

A 200 mL two-necked flask was charged with 120 mL of dimethylformamide (DMF) and the solvent was degassed with argon for 10 min.H_2_TClPP porphyrin (**1**) (200 mg, 1.0 eq) was added to the main reaction vessel and the mixture was heated to 160 °C under magnetic stirring. CoCl_2_.6H_2_O (69 mg, 2 eq) was added to the mixture under argon and the mixture was left at 160 °C and magnetic stirring for a further 2 h until the TLC analysis (SiO_2_, CHCl_3_ eluent) revealed no trace of the free-base porphyrin. The resulting mixture was cooled to 0 °C. The red-orange color precipitate was collected by filtration, washed with water then with *n*-hexane and air-dried. The obtained pure [Co^II^(TClPP)] complex (**2**) was obtained with a 91% yield (195 mg).

Anal (%) calcd for C_44_H_24_N_4_Cl_4_Co (809.45): C, 65.29; H, 2.99; N, 6.92; found: C, 65.17; H, 2.79; N, 7.10;UV-vis [CH_2_Cl_2_]: λ_max_ (ε.10^−3^ M^−1^.cm^−1^) 414(340), 532(56); ^1^H NMR (400 MHz, CDCl_3_) δ(ppm), 15.75 (s, 8 H β-pyrrole), 12.93 (s, 8H ArH), 9.9 (s,8H, ArH); FTIR (solid, $\bar{\nu }$, cm^−1^); 2966 ν[(CH) porphyrin], 1484 ν[(C=N), (C=C) porphyrin], 1007 [δ(CCH) porphyrin].

#### 2.3.3. Synthesis of [Co^III^(TClPP)Cl] (**3**)

[Co^II^(TClPP)] (**2**) (100 mg, 0.12 mmol) was dissolved in 10 mL of dichloromethane and 2 mL of HCl (27%) was added and the color changed from red orange to dark green. The mixture was stirred at 60 °C for 3 h. Then, the obtained precipitate was collected by filtration, washed with 25 mL of cold methanol, and dried under vacuum. The resulting cobalt(III) [Co^III^(TClPP)Cl] complex (**3**) was obtained with a yield of 78%.

Anal (%) calcd for C_44_H_24_N_4_Cl_5_Co (844.90): C, 62.55; H, 2.86; N, 6.63; found: C, 62.72; H, 2.98; N, 6.79; UV-vis [CH_2_Cl_2_]: λ_max_ (ε.10^−3^ M^−1^.cm^−1^) λ_max_: 442(296), 557(50), 596(36); ^1^H NMR (400 MHz, CDCl_3_): *δ* = 8.95 (s, 8 H β-pyrrole); 7.95(s, 8H ArH); 7.78(s, 8H ArH); FTIR (solid, $\bar{\nu }$, cm^−1^): 2900 ν[(CH) porphyrin], 1504 ν[(C=N), (C=C) porphyrin], 1060 [δ(CCH) porphyrin].

#### 2.3.4. Synthesis of [Co^III^(TClPP)Cl(NTC)]·CH_2_Cl_2_ (**4**)

The coordination compound **3** (20 mg, 0.02 (mmol) was mixed with nicotinoyl chloride hydrochloride C6H4ClNO·HCl (80 mg, 0.76 mmol) in 50 mL of dichloromethane at room temperature for 2 h. Blue-green crystals of [Co^III^(TClPP)Cl(NTC)]·CH_2_Cl_2_ (**4**) (NTC = nicotinoyl chloride) were obtained by slow diffusion of *n*-hexane into the dichloromethane solution (Yield ~85%).

Anal (%) calcd for C_51_H_30_N_5_Cl_6_OCo (1071.39): C, 57.17; H, 2.82; N, 6.54; found: C, 57.51; H, 2.69; N, 6.71; UV-vis [CH_2_Cl_2_]: λ_max_ (ε.10^−3^ M^−1^.cm^−1^): 455(335), 560(45), 696(59); ^1^H NMR (400 MHz, CDCl_3_): *δ* = 9.08 (s, 8 H β-pyrrole), 8.09 (s, 8H ArH), 7.75 (s, 8H ArH),7.28 (s, 3H ArH-ligand), 6.65 (s, ArH-ligand); FTIR (solid,  ν ¯, cm^−1^), 2950 ν[(CH) porphyrin], 1708 ν[(C=O) Ligand] 1496 ν[(C=N), (C=C) porphyrin], 1006 [δ(CCH) porphyrin].

## 3. Results and Discussion

### 3.1. IR and Proton NMR Spectroscopic Data

The IR spectra of the H_2_TClPP (**1**) free-base porphyrin and the [Co^II^(TClPP)] (**2**) and [Co^III^(TClPP)Cl] (**3**) starting materials are depicted in Appendix A, respectively.

Complex **4** exhibits an IR spectrum characteristic of a *meso*-arylporphyrin coordination compound and confirms the presence of the nicotinoyl chloride (NTC) axial ligand (Appendix A). Indeed, the ν(CH) stretching frequency values of the TClPP porphyrinate and the NTC axial ligand of **4** are in the 3050–2854 cm^−1^ range while the ν(C=C) and ν(C=N) stretching frequency values are 1496 cm^−1^. The δ(CCH) bending frequency value of **4** is shown at 1007 cm^−1^. The coordination of the NTC to the cobalt is confirmed by the absorption bands at 1708 cm^−1^ corresponding to the ν(C=O) stretching frequency. This value indicates that this axial ligand is coordinated to the central Co(III) metal through the pyridyl group and not the carbonyl group of the chloride acid.

^1^H NMR spectroscopy is a successful method to check whether a cobalt metalloporphyrin is a paramagnetic cobalt(II) complex or a diamagnetic cobalt(III) coordination complex with 3d^7^ and 3d^6^ fundamental state electronic configurations of the Co(II) and Co(III) cations, respectively. Cobalt(II) *meso*-arylporphyrin derivatives exhibit downfield-shifted ^1^H NMR spectra with β-pyrrole proton chemical shift values ~16 ppm and phenyl proton values in the range 13–9.8 ppm. On the other hand, for cobalt(III) *meso*-arylporphyrin complexes the β-pyrrole protons and of the phenyl ring are slightly downfield-shifted with respect to those of the related free-base porphyrins, with chemical shift values ~8.9 ppm and in the range 7.65–8.80 ppm, respectively. The NMR spectrum of complex **4** is depicted in Appendix A while the chemical shift values of the phenyl and the protons β-pyrrole of **1**–**4** and several related cobalt(II) and cobalt(III) metalloporphyrins are listed in Table 1. The β-pyrrole protons of **4** resonate at 9.08 ppm while the chemical shift values of the aryl protons of this complex are in the range 8.09–7.75 ppm, which confirms that our Co-TClPP-NTC derivative is a paramagnetic cobalt(III) metalloporphyrin. The protons of the NTC axial ligand resonate between 7.5 and 6.5 ppm.

### 3.2. Photophysical Properties

The UV-vis spectra of our four synthetic porphyrinic species (**1**–**4**) are depicted in Figure 1 while in Table 2 is reported the UV-vis data of these compounds and several similar porphyrin species.

The λ_max_ value of the Soret band of the free-base porphyrin H_2_TClPP is 421 nm while those of the Q bands are 522, 557, 599 and 651 nm which are characteristic for free-base *meso*-arylporphyrins. The insertion of cobalt leads to a reduction in the number bands Q from four to two and to a shift of the Soret band towards the blue at 414 nm. The absorption spectrum of [Co^III^(TClPP)Cl] (**3**) shows a red-shift Soret band at 442 nm, which is an indication of the oxidation of cobalt(II) to cobalt(III) (Table 2). In addition, the coordination of the nicotinoyl chloride (NTC) to the Co(III) center metal of (**3**) leads to an important red shift of the Soret band (455 nm) and the Q bands (560 and 696 nm). For this Co(III) pentacoordinated metalloporphyrin (**4**), the significant redshift of the Soret and Q bands explain the green color of this Co(III) metalloporphyrin, which is due to the important deformations of the porphyrin core (see the crystallographic section) reported by Weiss et al. [36].

The values of the optical gap (E_g_) of **1**–**4** were calculated by applying the Tauc relationship ${\left(\alpha h\right)}^{2}=A[h-E_{g}$], where *A* is a constant parameter depending on the transition probability, *hν* is the incident photon energy and *α* is the optical absorption coefficient extracted from absorbance data [35]. The intercept from the solid lines and the x-axis allows the determination of E_g_ values (Appendix A) of **1**–**4** which are 1.88, 2.01, 1.96 and 1.73 eV, respectively (Appendix A, Table 2). The free-base porphyrin H_2_TClPP (**1**) exhibits a E_g_ value which is lower than those of [Co^II^(Telp)] (**2**) and [Co^III^(TClPP)Cl] (**3**) which could be justify by the fact that the non-metalated porphyrin presents a higher flexibility of the porphyrin core leading to the destabilization of the HOMO-LUMO orbitals and therefore lowering the energy of the HOMO-LUMO orbitals. In the case of [Co^III^(TClPP)Cl(NTC)]·CH_2_Cl_2_ (**4**), the E_g_ energy (1.73 eV) is lower than that of the H_2_TClPP free-base porphyrin (1.88 eV), which is related to the very important distortion of the porphyrin macrocycle of this complex (see crystallographic section).

Porphyrins compounds usually present two emission transition types: (i) the S_1_ → S_0_ transition from the S_1_ first excited state to the S_0_ ground state corresponding to the Q bands [O(0,0 and Q(0,1)] and (ii) the S_2_→S_0_ transition from the second excited state S_2_ to the ground state S_0_ corresponding to the Soret band. Given that the S_2_→ S_0_ emission transition is very weak compared to the S_1_→S_0_ transition for porphyrin species, the former fluorescence is usually not considered for porphyrins and metalloporphyrins.

The fluorescence spectra of our four porphyrinic species (**1**–**4**) are presented in Figure 2 and the fluorescence parameters of these compounds and some free-base *meso*-arylporphyrins and Co(II) and Co(III) related metalloporphyrins are reported in Table 3.

As shown in Figure 2, the H_2_TClPP porphyrin (**1**) presents Q(0,0) and Q(0,1) emission band values at 650 and 714 nm, respectively. For the cobaltous [Co^II^(TClPP)] (**2**) complex, the O(0,0) and Q(0,1) bands are blue-shifted with λ_max_ values of 641 and 709 nm, respectively. Our two cobalt(III) complexes (**3**–**4**) show blue-shifted Q(0,0) and Q(0,1) bands at 637 and 699 nm, respectively. Generally, the fluorescence quantum yields (φ_f_) of free-base porphyrins are greater than those of the corresponding metalated, which is explained by the electron transfer from the donor part of the metalloporphyrin to its acceptor part. The donor part of a porphyrin complex is the porphyrin core, and the acceptor part is the central metal. Indeed, the φ_f_ value of **1**, which is 0.089, is higher than that of **2** with a value of 0.035. The fluorescence lifetime (τ_f_) values have usually the same trend than the fluorescence quantum yields. Thus, the τ_f_ of **1**–**2** are 7.40 and 6.40 ns, respectively. For the two Co(III) complexes [Co^III^(TClPP)Cl] (**3**) and [Co^III^(TClPP)Cl(NTC)]·CH_2_Cl_2_ (**4**), we noticed: (i) compounds **3**–**4** present very close fluorescence lifetime values (~2.40 ns), (ii) for **3**, the φ_f_ value is as expected smaller than that of the H_2_TClPP with a value of 0.051 and (iii) compound **4** exhibits high φ_f_ value (0.065). An explanation of this behavior could be the very important deformation of the porphyrin core, which has an important effect on the acceptor–donor characteristics of the [Co^III^(TClPP)Cl(NTC)] molecule.

### 3.3. X-ray Molecular Structure of Complex **4**

The molecular structure of our Co(III)-TClPP-Cl-NTC derivative (**4**) was determined by X-ray diffraction. Single crystals were obtained by slow diffusion of *n*-hexane through a dichloromethane solution containing complex **4**. The crystallographic data and the refinement of the structure of this species are presented in Appendix A while a selection of distances and angles of the same species are given in Appendix A. Complex **4** crystallizes in the monoclinic crystal system with *P2_1_*/*c* space group. One [Co^III^(TMPP)Cl(NTC)] molecule and one disordered dichloromethane solvent molecules are the only constituents of the asymmetric unit of **4**. The ORTEP drawing of this Co(III) metalloporphyrin is illustrated in Figure 3. For this Co(III)-NTC porphyrin derivative, the cobalt(III) central ion is coordinated by the four pyrrole N atoms of the porphyrin. The chloride and the nicotinoyl chloride (NTC) axial ligands occupy the two apical sites of the distorted square-bipyramidal coordination polyhedra.

Senge et al. [39] reported that the macrocycle of the porphyrin presents four ideal deformations: (i) the doming deformation (*dom*) due to the displacement of the metal atom out of the 24-atom mean plane of the porphyrin core and the displacement of the nitrogen atoms to the axial ligand, (ii) the ruffling distortion (*ruff*) is characterized by the high values of the displacement of the *meso*-carbon atoms over and below the porphyrin mean plane (Figure 4a,b), (iii) the saddle deformation (*sad*) originates from the displacement of the pyrrole rings alternatively above and below the mean porphyrin core so that the pyrrole nitrogen atoms are out of the mean plane, and (iv) the waving distortions (*wav*), for which the four fragments “β-carbon-α-carbon-*meso*-carbon-α-carbon-β-carbon” are alternatively below and above the mean plane of the porphyrin macrocycle.

The most structural important feature of our new cobalt(III) chloride-nicotinoyl chloride derivative (**4**) is the very important ruffling and waving deformations of the porphyrin core (Figure 4c). Thus, as shown in Figure 5, the displacements of the *meso*-carbons from the mean plane of the 24-atom porphyrin core are −67, 63, −63 and 64.10^−2^ Å, indicating a very important ruffling deformation. The four fragments “β-carbon-α-carbon-*meso*-carbon-α-carbon-β-carbon” also present very high displacements vis-à-vis the mean plane of the 24-atom porphyrin core with values “−26 −35 −67 −39 −30”, “+22 +37 +63 +36 +27”, “−23 −35 −63 −36 −32” and “+16 +36 +64 +42 +27” 10^−2^ Å, which confirms the significant waving deformation of the porphyrin core. It is noteworthy that the deformation of the porphyrin core and especially the ruffling distortion and the variation in the equatorial mean distance between the center metal and the nitrogen atoms of the porphyrin core (M^__^Np, M = center ion) are related. Thus, the greater the ruffling deformation is, the smaller the Co–Np distance is and vice versa [39].

Inspection of Table 4 shows that (i) cobalt(III) metalloporphyrins with significant ruffling and waving deformations present very short Co–Np distances, while those with small deformations of the porphyrin core show longer Co–Np distance values as just indicated above, and (ii) our Co(III)-Cl-NTC derivative (**4**) presents a very short Co–Np bond distance of 1.947(3) Å, which is in accordance with the fact that this species exhibits very important ruffling and waving deformations. Consequently, the Soret and Q bands of the spectrum of **4** and the Q bands of the fluorescence spectrum are very red-shifted.

The important ruffling and waving deformations of the porphyrin core of **4** are most probably due to coordination of the sterically hindered nicotinoyl chloride axial ligand to the Co(III) center ion. Thus, in order to minimize the interaction of this ligand with the phenyls or the TClPP porphyrinate, the porphyrin core is twisted severally so that the hydrogen atoms of the pyridyl group of the nicotinoyl chloride axial are as far as possible from the hydrogens of the phenyl closest to the porphyrin, leading to the formation of a “ligand binding pocket” in which the nicotinoyl chloride axial is located (Figure 4c).

There are obviously no classical hydrogen bonds in the structure of **4** for which the crystal stability is described as follows (Appendix A): (i) the chlorine Cl_2_ in *para* position of a phenyl group of one [Co^III^(TClPP)Cl(NTC)] molecule and the C8 atom of one pyrrole ring of a neighboring TClPP porphyrinate of a [Co^III^(TClPP)Cl(NTC)] complex are hydrogen bonded with a C8^__^H8**^…^**Cl_2_ distance of 3.398 (4) Å, (ii) the carbon atoms C31 and C32 of one phenyl ring of one [Co^III^(TClPP)Cl(NTC)] complex are H bonded to the centroid Cg4 of the N4/C16-C19 pyrrole ring and the Cl5 chloride axial ligand of the same [Co^III^(TClPP)Cl(NTC)] closest neighbor, respectively (Appendix A). The C31^__^H31**^…^**Cg4 and C32^__^H32**^…^**Cl5 distance values are 3.503(3) and 3.374 (3) Å, respectively, (iii) the carbon C47 of the phenyl ring of the nicotinoyl chloride (NTC) axial ligand is weakly linked to the centroid Cg3 of the N3/C11-C14 of a nearby [Co^III^(TClPP)Cl(NTC)] molecule with a C47^__^H47**^…^**Cg3 distance of 3.458(5) Å and (iv) the centroid Cg1 of the N1/C2-C4 of the later molecule is weakly H bonded to the carbons C51A and C51B of the disordered dichloromethane solvent molecule with a C50A^__^H51B**^…^**Cg1 and C50B^__^H51C**^…^**Cg1 distance values of 3.675(6) and 3.61(5) Å.

We noticed the presence of small voids 1.2 Å and a grid of 0.7 Å perpendicular to the [010] direction (Appendix A). These voids correspond to 2% of the cell volume with a total volume of 95.12 Å^3^ per cell.

### 3.4. Hirshfeld Surface Analysis

In order to further understand the intermolecular interactions in the crystal of compound **4**, Hirshfeld surface (HS) analysis was carried out by using Crystal Explorer 17.5 [47].

The white surfaces in the HS plotted over d_norm_, indicates contacts with distances equal to the sum of van der Waals radii while the red and blue colors indicate distances shorter or longer than the van der Waals radii, respectively [48]. As shown in Figure 6a, the red spots in the Hirshfeld surface presented by the dnorm surface correspond to the C8^__^H8**^…^**Cl2, C32^__^H32**^…^**Cl5, C31^__^H31**^…^**Cg4, C47^__^H47**^…^**Cg3, C51A^__^H51B**^…^**Cg1 and C51B^__^H51C**^…^**Cg1. As already indicated by the PLATON calculations [49], the most abundant contributor of the total Hirshfeld surface around complex **4** is the H…Cl/Cl…H interaction which contributed by 33.2%. The other noticeable contributors are: H…H (29.7%), C…Cl/Cl…C (7.4%), Cl-Cl (5.1%) H…O/O…H (3.6%) (Figure 7).

Both the curvature and the shape indices can also be utilized to make identifications of typical stacking modes and the manners in which closely spaced molecules engage with each other. The shape index for **4** shows a red concave area on the surface around the acceptor atom and a blue area around the donor H-atom (Figure 6b) [50,51]. The curvedness is a function of the root mean square curvature of the surface and the maps of curvedness on the HS for complex **4** indicate no plain surface patches, indicating that there is no stacking interaction between the molecules (Figure 6c).

### 3.5. Cyclic Voltammetry Investigation on [Co^III^(TClPP)Cl(NTC)]·CH_2_Cl_2_ (**4**)

The electrochemical properties of **4** were assessed by cyclic voltammetry with tetra-n-butylammonium hexafluorophosphate (TBAPF_6_) used as supporting electrolyte (0.1 M) in dichloromethane under an argon atmosphere. The voltammogram of this cobalt(III) metalloporphyrin is illustrated by Figure 8.

The reduction in [Co^III^(TClPP)Cl(NTC)]·CH_2_Cl_2_ (**4**) has uncoupled cathodic and anodic peaks that are tagged **1a** (reduction) and **1b** (oxidation) (Figure 2) and were spectro-electrochemically analyzed by Kadish et al. [52] for the [Co^III^(TPP)Cl] (TPP = *meso*-tetraphenylporphyrinate) coordination compound as inlaying a Co(III)/Co(II) reduction. In the case of [Co^III^(TPP)Cl], the reduction (**1a**) occurs at E_cp_ = −0.20 V (E_cp_ = cathodic potential) and the reoxidation of the generated Co(II) complex (peak **1b**) occurs at E_ap_ = 0.57 V (E_ap_ = anodic potential) (Figure 8, Table 5). For complex **4**, the E_cp_ (reduction **1a**) and E_ap_ (oxidation **1b**) values are −0.37 V and −0.75 V, respectively (Figure 9 Table 5). The large difference in the Ecp and E_ap_ for the Co(III)-Cl-TPP and the Co(III)-Cl-NTC-TClPP could be caused by the important deformation of the porphyrin core in the case of complex **4** leading to the narrowing the energy levels of the HOMO and LUMO orbitals of this species. The uncoupled Co(III)/Co(II) reactions of cobalt(III) porphyrin complexes were explained [52] by the “box mechanism” shown in Figure 2. In this mechanism, the reduction and reoxidation of the [Co^III^(TPP)Cl] occur via two distinct reversible electron-transfer steps, each a chemical reaction:

As reported by Kadish et al. [52], the second reduction of [Co^III^(TPP)Cl] occurs at E_cp_ = −0.90 V [process (3) in Figure 8] involving Co(II)/Co(I) reduction and indicates the characteristics of a couple of chemical reactions (Figure 2). For our Co(III)-Cl-NTC-TClPP derivative, the E_cp_ value for the second reduction occurs at −1.29 V, which is shifted toward the negative potential, due probably to the significant deformation of porphyrin core of **4**. The third reduction of [Co^III^(TPPCl] reported by Kadish et al. involves the ring reduction of the electrogenerated [Co^I^(TPP)(CH_2_Cl] with E_cp_ value of −1.42 V. For complex **4**, a similar third reduction occurs at E_cp_ = −1.65 V. Based on the scheme proposed by Kadish et al. for the second reduction of [Co^III^(TPP)Cl] (Figure 3), is proposed to describe the second reduction of our Co(III) porphyrinic complex **4**:

The anodic part of the cyclic voltammogram of **4** contains two reversible one-electron oxidations assigned to the oxidation of the porphyrin ring where the E_1/2_ values are 1.06 and 1.43 V, respectively. These values are shifted to more positive potential than those of [Co^III^(TPP)Cl] (0.90 and 1.15 V, respectively) which also could be attributed to the significant deformation of the porphyrin macrocycle of **4**.

The photodegradation reactions of the malachite green (MG) dye using **1**–**4** as catalysts were monitored by UV spectroscopy. The λ_max_ of the absorption band of the MG at 618 nm was utilized to make an approximate assessment of the decolorization rate of the organic dye. The variation of λ_max_ of the absorption of MG dye upon radiation time, using **1**–**4** is reported in Appendix A. As shown in this figure, the degradation of MG dye did not occur without adding compounds **1**–**4**.

The effective degradation of MG dye (C_t_/C_o_ versus time curve) using **1**–**4** as photocatalysts indicates that [Co^III^(TClPP)Cl(NTC)]·CH_2_Cl_2_ (**4**) shows the higher degradation efficiency after 60 min of irradiation with a yield value of 95% (Figure 9). For compounds **1**–**3**, the degradation yields are 90%, 80% and 84%, respectively.

The photocatalytic discoloration of a dye is believed to take place according to the following mechanism. When a catalyst is exposed to UV radiation, electrons are promoted from the valence band to the conduction band. As a result, an electron–hole pair is produced [53], and $e_{cb}^{-}$ and $h_{vb}^{+}$ are the electrons in the conduction band and the electron vacancy (holes) in the valence band, respectively. Both these entities can migrate to the catalyst surface, where they can enter a redox reaction with other species present on the surface. In most cases, $h_{vb}^{+}$ can react easily with surface bound H_2_O to produce ^•^OH radicals, whereas $e_{cb}^{-}$ can react with O_2_ to produce superoxide radical anions of oxygen [54] (Figure 4). This reaction prevents the combination of the electron and the hole that are produced in the first step. The ^•^OH and O_2_^•^ produced in the above manner can then react with the dye to form other species and is thus responsible for the discoloration of the dye.

The reusability of the photocatalysts was considered. Thus, our four porphyrinic derivatives (**1**–**4**) used as catalysts were separated by filtration, washed with distilled water after each run, then dried and further subjected to subsequent runs under the same conditions. Appendix A indicates that the regeneration process could be repeated for 4 cycles, without appreciable activity loss. The regenerated catalysts were also characterized by FTIR analyses after each cycle and no change was observed.

The excellent photocatalytic degradation yields, and the cyclic stable performance testified that **1**–**4** could be candidates for other photocatalysis reactions.

## 4. Photovoltaic Performance of DSSCs

In the present work, the devices consisted of the porphyrin complexes **3** and **4** located between two electrodes, the “ITO-coated glass” and aluminum. The current vs voltage plots of the ITO/Pm/Al systems (Pm = [Co^III^(TClPP)Cl] or [Co^III^(TClPP)Cl(NTC)]·CH_2_Cl_2_]) is depicted in Figure 10. The displayed AFM image clearly reveals the formation of a homogeneous film with a great layer structure (1.89 nm).

This investigation makes it possible to determine the transport properties in the organic materials. Current–voltage plots (J-V) of compounds **3** and **4** were recorded in the dark and at room temperature.

As shown by Figure 10, the (J-V) plots are nonlinear, while the ITO/Pm/Al devices exhibit a clear moderate rectifying performance due to the injection of ITO charges for complexes **3** and **4**, leading to an organic device that may be a photovoltaic device [55,56]. The threshold voltages of complexes **3** and **4** are 0.592 and 0.652 V, respectively.

By studying the current-voltage (J-V) curves, we can observe the existence of two different regimes, which depend on the applied voltage.

At low voltages, the first regime presents a symmetry characteristic explained by the localized state with defects causing localized gap states. The second regime exhibits an asymmetric characteristic. This is related to the injection process of the electron and hole barriers due to the difference in the work functions of the two electrodes.

Better analysis of (J-V) characteristics based on semi-logarithmic representation is presented in Figure 11. Two regions have been noticed on these curves, the first region is linear; the current being limited by the resistance via the shunt resistance R_sh_ [57,58,59]. In the second region, the current starts to saturate due to the series resistance R_s_.

The J_s_ values of the developed devices were obtained from the experimental (J-V) data, and $\Phi_{b}$ was calculated following the following equation (Equation (1)):(1)$$J_{s}=\mathrm{SS}*\;\exp\left(\frac{-{q\Phi }_{b}}{\mathrm{KT}}\right)$$
where S is the constant area and S* is the Richardson constant, k the Boltzmann constant (1.38 × 10^−23^ J K^−1^), T the temperature and q the electronic charge.

Comparing the parameters of the two complexes, we concluded that complex **3** has a minimum barrier height value compared to complex 4 (Table 6). This can be explained by the presence of the ligand, which affects the charge transport mechanism.

Following the insertion of the carrier charges, their transport across the active surface to the opposite electrodes is defined by the conduction characteristics. The J–V characteristics in log-log plot of the compounds **3** and **4** in dark conditions are shown in Figure 12. The investigation of these plots suggests that the dependency of the current on the applied voltage seems to follow the power law: J = Vm. The current density is defined by the space charge of the carriers injected by the electrodes. By observing the plot, we have noticed the existence of two regimes. These regimes depend on the value of the slope m.

For the first phase, the current depends on the voltage (m = 1), which defines an ohmic region, due to the presence of a quantity of interface barrier preventing charge injection.

The current density is calculated by the following equation (Equation (2)):(2)$$J_{\Omega }={\mathrm{qp}}_{0}\mu \frac{V}{d}$$
where μ is the charge carrier mobility, q is the electron charge and d is the film thickness.

In the second phase, the voltage increases, and the current depends on the voltage (m = 2), corresponding to the space charge limited conduction (SCLC) mode [60,61,62,63]. In this phase, the current density is given by the following formula (Equation (3)):(3)$$J_{\mathrm{SCLC}}=\frac{9}{8}{\varepsilon \mu }_{\mathrm{eff}}\frac{V^{2}}{d^{3}}$$
where $\mu_{\mathrm{eff}}$ is the effective carrier mobility and ε is the permittivity of the material.

## 5. Conclusions

In summary, the synthesis and spectroscopic characterization of the free-base porphyrin H_2_TClPP (**1**), the [Co^II^(TClPP)] (**2**)the [Co^III^(TClPP)Cl] (**3**) starting materials and the [Co^III^(TClPP)Cl(NTC)]·CH_2_Cl_2_ (**4**) complex (NTC = nicotinyl chloride) were performed. The most important feature of the X-ray molecular structure of **4** is the very important ruffling and waving deformations of the porphyrin core. The Hirshfeld surfaces analysis on this later cobalt(III) metalloporphyrin shows that the crystal lattice is principally supported by C^__^H**^…^**Cl and C^__^H**^…^**Cg (Cg is the centroid of a pyrrole ring) involving both [Co^III^(TClPP)Cl(NTC)] and the dichloroethane solvent molecules. The UV-vis and fluorescence spectra of **4** are red-shifted due to the significant deformation of the porphyrin macrocycle. The electrochemical property of **4** was investigated using cyclic voltammetry, showing that the reduction potentials are shifted to negative values while the oxidation potentials are shifted to positive values compared to [Co^III^(TPP)Cl]. This feature is most probably due to the significant deformation of the porphyrin ring of **4**. Furthermore, the photodegradation of the malachite green dye, using **1**–**4** as catalysts, gives good yields between 80% for **2** and 95% for compound **4**. The photovoltaic characteristics of DSSCs based on complex **3** and **4** were measured.

In summary, we successfully prepared a new hexacoordinated cobalt(III) *meso*-arylporphyrin complex with the chlorido and the nicotinoylchloride axial ligands with the formula [Co^III^(TClPP)Cl(NTC)]·CH_2_Cl_2_ (TClPP = the *meso*-tetra(*para*-chlorophenyl)porphyrinate and NTC is the nicotinoylchloride axial ligand) (**4**). This Co^3+^ metalloporphyrin (**4**) and the H_2_TCPP (**1**), [Co^II^(TClPP)] (**2**) and [Co^III^(TClPP)Cl] (**3**) materials were characterized by UV-vis, IR, ^1^H NMR, fluorescence, mass spectrometry. Complex **4** was also characterized by single-crystal X-ray molecular structure and cyclic voltammetry. The most important feather of **4** is the very important deformation of its porphyrin core. This has consequences on several proprieties of complex **4**: (i) a very red shift of the UV-vis spectrum, (ii) a high φ_f_ value (0.065) where the very important deformation of the porphyrin core effects the acceptor–donor characteristics of this Co(III) complex, and (iii) the E_1/2_ values of the two reversible oxidation waves of the porphyrin ring are shifted to the positive potentials. All four porphyrinic compounds (**1**–**4**) were tested as catalysts in the photochemical degradation of malachite green (MG) dye, where complex **4** gave the best degradation yield (95%). Furthermore, the electrical properties of complexes **3**–**4** obtained based on their current voltage curves show that the nature of the axial ligand affects the performance of the cells including these two cobalt(III) metalloporphyrins.

## Data Availability

Not applicable.

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
