# Peer review of "Synthesis of New Cobalt(III) Meso-Porphyrin Complex, Photochemical, X-ray Diffraction, and Electrical Properties for Photovoltaic Cells"

_molecules, 2022, doi:10.3390/molecules27248866_

Round 1
Reviewer 1 Report
1. Please remove the Table 1 and Table 2 into ESI.
2. Please do all the IR data for the four complexes.
3. The manuscript contains spelling/grammatical errors. So, the language should be polished thoroughly, such as Eg should be changed into Eg. 4. EPR is not characterized used to explain the photocatalysis mechanism, there is no EIS and other tests further explain the mechanism; 5. The paper mainly reports photocatalytic degradation of pollutants, but there is no degradation rate;6. Please provide the XPS, which will be confirmed the Chemical valence.
7. Please trim the conclusion section, it is so trivial.
8. Some updated refs should be compared and cited, such as “The majority of organic dyes are toxic and difficult to biode-41 grade [17,18]. Among methods used for the decolorization of dyes, the heterogeneous photocatalysis is one of most used.”, the related refs could be cited, J. Catal.,2021, 394, 397-405; Appl. Catal. B: Environ. 2022, 306, 121095; J Colloid Interface Sci , 2022, 607, 281; New J. Chem., 2022, 46, 19577–19592; CrystEngComm, 2022, 24, 6933–6943 and Mater. Today. Commum., 2022, 31,103514.
9. Please provide the Scheme on the photocatalysis mechanism.
Reviewer 2 Report
The structural characterisation and spectroscopic data are sound. However, the novelty is quite limited and only one new compound is reported. The Co(III) porphyrin derivative is extensively studied in this article but I am not fully convinced if a broad enough readership will be attracted. The manuscript is also quite lengthy.
I suggest to significantly shorten the manuscript, i.e. by at least doing the following:
Move Table 1 to supporting info (SI).
Shorten Table 2 or add some crucial structural properties to Fig. 5.
Move Figs. 1 and 2 to SI
Combine Figs. 6 to 8 into one Figure.
Move Figs. 9 and 1o to SI, shorten the text accordingly
Move Fig. 15 to SI
From the experimental section, move General Experimental Information to SI (particularly 2.1-2.4)
2.5: Which standard electrode was used? I assume Ag/AgNO3? Please specify
Once the manuscript is significantly shortened and reduced in volume it should be ready for publication in Molecules
Round 2
Reviewer 1 Report
The authors have addressed all the comments.